# Evaluating Neuroprotective Effects of Uridine, Erythropoietin, and Therapeutic Hypothermia in a Ferret Model of Inflammation-Sensitized Hypoxic-Ischemic Encephalopathy

**DOI:** 10.3390/ijms22189841

**Published:** 2021-09-11

**Authors:** Kylie A. Corry, Olivia R. White, AnnaMarie E. Shearlock, Daniel H. Moralejo, Janessa B. Law, Jessica M. Snyder, Sandra E. Juul, Thomas R. Wood

**Affiliations:** 1Division of Neonatology, Department of Pediatrics, University of Washington, Seattle, WA 98195, USA; kcorry@uw.edu (K.A.C.); whiteoli@uw.edu (O.R.W.); ashearl@uw.edu (A.E.S.); moralejo@uw.edu (D.H.M.); janessal@uw.edu (J.B.L.); sjuul@uw.edu (S.E.J.); 2Department of Comparative Medicine, University of Washington, Seattle, WA 98195, USA; snyderjm@uw.edu; 3Center on Human Development and Disability, University of Washington, Seattle, WA 98195, USA

**Keywords:** neonatal, asphyxia, therapeutic hypothermia (TH), erythropoietin, uridine

## Abstract

Perinatal hypoxic-ischemic (HI) brain injury, often in conjunction with an inflammatory insult, is the most common cause of death or disability in neonates. Therapeutic hypothermia (TH) is the standard of care for HI encephalopathy in term and near-term infants. However, TH may not always be available or efficacious, creating a need for novel or adjunctive neurotherapeutics. Using a near-term model of inflammation-sensitized HI brain injury in postnatal day (P) 17 ferrets, animals were randomized to either the control group (*n* = 43) or the HI-exposed groups: saline vehicle (Veh; *n* = 42), Ur (uridine monophosphate, *n* = 23), Epo (erythropoietin, *n* = 26), or TH (*n* = 24) to test their respective therapeutic effects. Motor development was assessed from P21 to P42 followed by analysis of cortical anatomy, ex vivo MRI, and neuropathology. HI animals took longer to complete the motor assessments compared to controls, which was exacerbated in the Ur group. Injury resulted in thinned white matter tracts and narrowed cortical sulci and gyri, which was mitigated in Epo-treated animals in addition to normalization of cortical neuropathology scores to control levels. TH and Epo treatment also resulted in region-specific improvements in diffusion parameters on ex vivo MRI; however, TH was not robustly neuroprotective in any behavioral or neuropathological outcome measures. Overall, Ur and TH did not provide meaningful neuroprotection after inflammation-sensitized HI brain injury in the ferret, and Ur appeared to worsen outcomes. By comparison, Epo appears to provide significant, though not complete, neuroprotection in this model.

## 1. Introduction

Perinatal insults, such as inflammation and birth asphyxia, often contribute to both preterm and term brain injury [1]. Hypoxic-ischemic encephalopathy (HIE) occurs in 1–4 per 1000 live births in high-income countries, and around 12 per 1000 live births in low- and middle-income countries [2]. The incidence of HIE in preterm populations is likely underestimated because of its lack of a clear clinical definition, but a retrospective report of infants born at less than 37 weeks’ gestation in the United States between 2008 and 2011 showed an incidence of moderate to severe HIE of 37.3 per 1000 live births [3]. TH has been shown to significantly improve the outcomes of term and near-term infants with HIE in high resource settings, with a number needed to treat of 7 [4]. TH has not been proven to be beneficial in term infants in lower resource settings [5], or in preterm infants, so new approaches to neuroprotection are needed for these populations.

Animal models that accurately and reproducibly reflect human postnatal brain development as well as the pathophysiology and sequelae of HIE are also needed to test potential neurotherapeutics. For instance, few gyrencephalic or large animal models exist that allow for long-term assessment of injury to the developing cortex and white matter, which are particularly vulnerable to inflammation and HI in both preterm and term infants [6]. The ferret is an ideal animal model to examine the long-term effects of injury on brain development and response to treatment in prematurity and asphyxia-related disease states [7,8]. Postnatal ferret brain development is histologically similar to the rapid brain maturation that occurs in the second half of human gestation [9,10]. Our group recently published a protocol utilizing lipopolysaccharide (LPS) inflammation presensitized hypoxia-ischemia plus hyperoxia (HIH) to model hypoxic-ischemic (HI) brain injury in the near-term (P17) ferret [11]. The P17 ferret brain developmentally correlates to an infant at 32–36 week’s gestation [12].

The present study presents two cohorts: an initial cohort testing a putative neuroprotectant, uridine monophosphate, with a later cohort testing two established neurotherapeutics-erythropoietin (Epo) and TH. We hypothesized that Ur and Epo would provide significant neuroprotection against inflammation-sensitized, hypoxic-ischemic brain injury, but that LPS presensitization would negate the protective effect of TH in this model [13].

## 2. Results

### 2.1. Circulating Uridine and Uracil Levels

Baseline concentrations of uracil were 10× higher on average than those of Ur (Appendix A). Circulating Ur and uracil levels after s.c. administration of 330 mg/kg peaked at 30 min, with uracil levels approximately double that of Ur. After administration of 1000 mg/kg, Ur levels peaked at around 100,000 ng/mL within 30 min and remained elevated up to the 120 min time point. The concentration of uracil increased more slowly at the higher dose and peaked at around 90,000 ng/mL at 2 h. Due to the suggestion of saturation kinetics at 1000 mg/kg, and based on previous studies targeting a circulating neuroprotective dose of Ur around 10,000 ng/mL [14], animals in the Ur treatment group were administered 500 mg/kg every 12 h for a total of five doses.

### 2.2. Model Outcomes

The first cohort consisted of 71 kits, 59 of which survived the HIH insult and were randomized to littermate control, Veh, and Ur treatment groups: *n* = 19, *n* = 17, and *n* = 23, respectively. The second cohort consisted of 115 kits, 99 of which survived the HIH insult with *n* = 24, *n* = 25, *n* = 26, and *n* = 24 randomized to littermate control, Veh, Epo, and TH, respectively. After the initiation of treatment, and across both cohorts, mortality was 11.9% in the Veh group, 12.5% in the TH group, 7.7% in the Epo group, and 34.8% in the Ur group. This resulted in *n* = 43, *n* = 37, *n* = 15, *n* = 24, and *n* = 21 survivors to P42 in the control, Veh, Ur, Epo, and TH groups, respectively. An outline of the experimental outcomes, groups, and survival to assessment has been provided in Appendix A.

All HIH animals lost weight immediately after HIH exposure at P17. Males lost a median (IQR) weight of 10 g (3–17 g) body weight and females lost a median weight of 11 g (3–16 g) (Figure 1A,B). In contrast, control males gained a median weight of 11 g (5–17 g) and control females gained a median weight of 10 g (6–14 g). After the initial weight loss, the rate of weight gain was similar between groups, resulting in a significant difference between HIH animals and controls (Figure 1C).

### 2.3. Behavioral Testing

#### 2.3.1. Early Reflex Testing

HIH animals were non-significantly slower at completing all reflex tests (NG, CA, RR, and total time) when compared to the control animals. Analysis of the reflex AUCs (median; IQR) showed that Ur animals (431.8 s · days; 309.0–486.4 s · days) took longer to back away from the edge of the testing field in CA compared to control animals (361.7 s · days; 296.4–444.0 s · days), but significance was lost after adjusting for multiple comparisons (data not shown). Additionally, the Ur animals (649.5 s · days; 630.8–662.5 s · days) took significantly longer to take a full step 91° away from the CA starting position compared to control animals (634.5 s · days; 605.3–653.8 s · days) (*p* = 0.02; data not shown). Throughout testing RR, the Veh (456.8 s · days; 374.4–539.6 s · days) and Ur (456.3 s · days; 354.3–543.8 s · days) animals took significantly longer to take a full step following righting than control animals (414.4 s · days; 349.1–485.0 s · days) (Veh: *p* = 0.021, Ur: *p* = 0.014; data not shown). Throughout the testing period, Ur animals (8263.3 s · days; 7782.8–8658.0 s · days) had a longer total time to complete all tests when compared to controls (8083.3 s · days; 7690.5–8486.2 s · days) (*p* = 0.028; Figure 2A). This behavioral deficit was consistent in the Veh and TH animals but was not statistically significant.

#### 2.3.2. Open Field

Compared to controls, Ur animals visited the corners 46% more often and were 51% more inactive (*p* = 0.047 and *p* = 0.034, respectively; Figure 2B,C). Veh animals visited the corners 23% more frequently and were 15% more inactive than controls. The TH and Epo groups had considerably smaller, non-significant increases from the control medians for corner frequency and time spent not moving (Figure 2B,C).

#### 2.3.3. Gait Analysis

All HIH groups had a wider forelimb base of support than control animals. However, significance only remained in the TH group after adjusting for multiple comparisons (*p* = 0.0189, Appendix A). All HIH-exposed groups had a wider hindlimb base of support as well (Ur: *p* = 0.011, TH: *p* = 0.0029, Epo: *p* = 0.0024; Figure 2D), though in the Veh group this difference was not statistically significant. Compared to control animals, animals in the Veh, Epo, and TH groups had 8.4, 4.8, and 9.5% increases in median foreprint area, respectively, but these differences were not significantly different (Figure 2E). By comparison, median foreprint area of the Ur group was 35.6% greater than that of the control group (*p* = 0.0007). In the Veh and Ur groups, phase lag between the left front and right hind paws was increased, but significance was lost after considering multiple groups (Appendix A). Similar to the pattern seen with foreprint area, compared to control animals, animals in the Veh, Epo, and TH groups had 13.0, 15.9, and 17.0% increases in median hindprint intensity, respectively, but these differences were not significantly different. Hindprint intensity of the Ur group was 23.9% greater than that of the control group (*p* = 0.0417; Appendix A). 

Compared to the number of attempted runs in the CatWalk in the control group (median of 6), the median number of attempted runs for Ur animals was 8 (*p* = 0.0361; Figure 2F). Ur animals also showed a significantly higher ratio of attempted to compliant runs, or failed runs, compared to controls (*p* = 0.0021). A similar pattern was observed for other HIH-exposed animals but none of these groups significantly differed from control animals.

#### 2.3.4. Morris Water Maze

Unlike rats, ferrets were not motivated to find the platform during the MWM trials, with only 7.5% success in the control group, and animals appeared comfortable swimming during the allotted time period. Movement characteristics during swimming were therefore analyzed rather than platform-mounting success (Appendix A). Compared to control animals, all HIH animals appeared to spend a greater period of time not actively swimming (data not shown), though significance was lost for the Veh group after adjusting for multiple comparisons. After adjusting for sex and trial number, logistic regression revealed that Epo animals were significantly less likely to show swim behavior similar to that of control animals. The Epo group had a significantly reduced odds (OR, 95% CI) of covering more than 50% of the water area (0.25, 0.07–0.90; Appendix A) or displaying circuitous swimming open water (0.32, 0.10–0.98; Appendix A). A similar trend of change in swim behavior was seen in Veh animals for both parameters, but these were not significant. Swim behavior of Ur and TH animals did not significantly differ from controls based on the swim parameters assessed.

### 2.4. Cortical Neuropathology Scores

When evaluating cortical neuropathology, animals that died due to acute brain injury received a score of 8.5 and were included in the analyses (Figure 3A). There was a significant increase in cortical neuropathology score in the Veh, Ur, and TH groups compared to controls (median scores of 1 for Veh and 1.5 for Ur and TH groups). Epo animals had cortical neuropathology scores that were more similar to the control group (median injury score of 0.0 for both).

### 2.5. Brain Measurements

The summed sulci of all HIH groups were significantly shorter than the control group (Veh: *p* < 0.0001, Ur: *p* = 0.012, Epo: *p* = 0.0018, TH: *p* < 0.0001; Figure 3B). There was a slight improvement in the summed sulci of the Ur animals compared to other HIH-exposed animals, but this difference was not significant. When the summed totals of all gyral widths were assessed, Veh, Ur and TH animals had a significant decrease in the overall gyral width when compared to the control animals (Veh: *p* < 0.0001, Ur: *p* = 0.0003, TH: *p* = 0.0002; Figure 3C). The Epo group had a significant improvement in the summed totals of all gyri widths when compared to the vehicle group (*p* = 0.012).

### 2.6. MRI

Relative to the control group, the Veh animals had lower FA values throughout the cerebral white matter and corpus callosum (85% CI, Figure 4A) with the greatest difference in the dorsal striatum of the subcortical basal ganglia (specifically the ventral pallidum, putamen, and corpus striatum) as well as in the peri-thalamic substantia innominata (95% CI, Figure 4A). Animals in the TH group had higher FA values than the Veh group throughout the cerebrum including the premotor and prefrontal cortices (CI 0.85, Figure 4B) as well as the corpus callosum and anterior and posterior cerebral white matter (CI 0.95, Figure 4B). The Epo group had higher FA values than the Veh group primarily in the anterior cerebral white matter, the anterior interior capsule, and in the adjacent olfactory nucleus (CI 0.85, Figure 4C). However, FA values in the hippocampus and peri-hippocampal cerebral white matter tracts were lower in the Epo group than in the control group (85 and 95% CI, Figure 4D). There were no statistically significant differences in FA values between the TH and Epo groups. Of note, there were no areas in which the Veh FA values were significantly greater than the control, TH, or Epo groups.

### 2.7. IHC and Quantitative IHC

No significant differences were observed in the MBP-stained ROIs across all groups. Iba-1 staining was significantly increased in the CC of the Veh group when compared to the control group (*p* = 0.013, Appendix A). Iba-1 was also increased in the CC of the TH group when compared to controls, but lost significance after multiple comparisons (*p* = 0.07; data not shown). The Veh group also displayed a non-significant increase in Iba-1 staining of the SWM, but this difference was not observed in the other HIH groups. The Epo and TH groups had increased GFAP staining in the thalamus (Epo: *p* < 0.0001; TH: *p* < 0.0001), CC (Epo: *p* = 0.0005; TH: *p* = 0.004), and SWM (Epo: *p* = 0.0012; TH: *p* = 0.0012) when compared to the control group. The Veh group also had an increased intensity of GFAP staining in the thalamus (*p* < 0.0001) and SWM (*p* = 0.0002) when compared to the control group, but there were no observable differences between the Ur and control groups in GFAP staining across all ROIs (Appendix A). The Veh, Epo, and TH groups had decreased Olig2 staining intensity in the CC and SWM when compared to the control group, but these differences were not statistically significant (data not shown). The patterns of white matter thinning in the animals exposed to the HIH insult were independently consistent in the three white matter ROIs inferomedial to the lateral sulci (Figure 5A), suprasylvian sulci, and pseudosylvian sulci when compared to controls. The sum of the other three white matter ROIs at the base of consecutive sulci exhibited white matter thicknesses that were significantly narrower in the Veh, Ur, and TH groups when compared to the control group (Veh: *p* < 0.0001, Ur: *p* < 0.0001, TH: *p* = 0.0002; Figure 5B). All HIH animals had significantly narrower CC measurements when compared to the control animals (*p* < 0.0001; Figure 5C). There was an improvement in the summed white matter thickness of the Epo group when compared to the Veh group, but this difference was not statistically significant (*p* = 0.054). 

Compared to the controls, the Veh animals developed significantly less myelination in the cortex (*p* = 0.037; data not shown). While Epo animals had cortical myelination that was more similar to control animals, TH animals developed less myelinated cortexes than control animals. However, this difference was not statistically significant.

## 3. Discussion

Neonatal encephalopathy in both preterm and term infants is often associated with hypoxia-ischemia, and is often initiated or exacerbated by exposure to infection or other sources of inflammation [15]. The ferret provides an exciting, highly translatable animal model to study neonatal encephalopathies. As the smallest gyrencephalic mammal, the ferret is born altricial and begins to develop cortical folds in the first 6–8 days of postnatal life [16]. This feature allows for the study of injury patterns that impede normal cortical development. We have previously described a reproducible injury protocol to model neonatal encephalopathies in the P17 ferret, when the brain is approximately equivalent to human brain at 32–36 weeks of gestation. Utilizing this model, we sought to test both known and potential neurotherapeutics. After LPS-sensitized HIH in the P17 ferret, neither Ur nor TH were robustly beneficial, though TH was associated with preservation of certain white matter areas on DTI compared to Veh-treated injured controls. By comparison, Ur may have contributed to an increase in mortality and worse performance during reflex development. In contrast, Epo was partially neuroprotective, particularly with respect to white matter and cortical structure, which is consistent with multiple other preclinical studies [17,18,19,20,21,22]. This suggests that the ferret HIH model responds as expected to established neuroprotective agents in the setting of LPS-sensitized HI brain injury, providing a robust gyrencephalic, large animal model of neonatal encephalopathies amenable to multiple long-term behavioral and neuropathological assessments. 

Uridine represents a potential therapy that would be particularly advantageous in low resource countries, as it is inexpensive and shelf-stable. When administered at a dose of 500 mg/kg for three consecutive days, Ur provided modest neuroprotection in a rat model of HIE [14,23]. The neuroprotective effects of Ur administration in rodents seems to be mediated through its antiapoptotic [14] and antioxidant actions [24]. However, despite being administered at a similar dose, Ur was not neuroprotective in this model and may have contributed to the increase in mortality. Even with the increased mortality in Ur treated animals, which likely removed the most severely injured animals, Ur treatment still delayed the acquisition of basic reflexes, decreased exploratory behavior in the open field and MWM arena, and increased brain pathology beyond that incurred from the HIH insult alone. Initially, there was a concern that exogenous Ur administration could result in a higher death rate due to altered thermoregulation, as high doses of Ur administered in human and rabbit experiments have been shown to induce hypothermia [25]; however, rectal temperatures in our ferret model were not affected by Ur treatment (data not shown). Additionally, metabolism of Ur by ferrets differs from humans, in which Ur is the primary circulating pyrimidine nucleoside. By comparison, the ferret appears to convert the majority of circulating Ur to uracil, unless given in high doses. Therefore, the lack of neuroprotection conferred by administering Ur in this model does not preclude its potential success in other HI models or modalities of brain injury. 

Epo is an erythropoietic hormone produced endogenously in response to chronic hypoxic events, including hypoxia and fetal stress in utero [26,27,28,29,30,31]. After acute neonatal brain injury, treatment with high doses of Epo confers neuroprotection acutely through antiapoptotic and anti-inflammatory actions in multiple animal models [17,18], as well as through longer acting protective mechanisms including angiogenesis, neurogenesis, and oligodendrogenesis [19,20,21,22]. Treatment with Epo in this model resulted in quicker weight recovery after HIH, with Epo-treated animals showing a growth rate more similar to controls than to the other HIH-exposed animals. There was also a trend across the late behavioral tests of the Epo animals behaving more similarly to controls than Veh, Ur, or TH treated animals, though this was rarely significantly different. However, interestingly, a separate swim pattern analysis found that compared to Veh and TH animals, Epo animals were less likely to have swim patterns similar to that of control animals. Understanding the variable effects of Epo on behavior is outside the scope of this paper but will be an important avenue for further investigation. Although Epo treatment was unable to significantly recover locomotor abilities after HIH, partial neuroprotection was observed in a number of neuropathological and morphological assessments. For example, animals in the Epo group had significantly improved cortical pathology scores and normalization of total gyral widths. Epo-treated animals also had significantly thicker white matter tracts in their SWM when compared to the Veh and Ur brains, but they still displayed significantly thinner white matter tracts in the SWM and CC regions when compared to the control brains. Similarly, although animals in the Epo treatment group had evidence of greater peri-hippocampal white matter injury than control animals by DTI, Epo-treated animals had DTI markers consistent with improved white matter preservation relative to the Veh group, particularly in the internal capsule. This suggests that the Epo treatment had localized effects on white matter protection and maturation following HIH, with region-specific responses to injury and therapy. It also aligns with some preclinical Epo studies in which Epo has not consistently been neuroprotective depending on model, dosing strategy, and outcome measures [32,33,34,35]. An important component of our future work in this model will be determining whether combinations of therapies are necessary to provide global neuroprotection after neonatal HI brain injury. 

The benefit of administering TH within 6 h of an HI event has been demonstrated through meta-reviews and large clinical trials [36], and we aimed to replicate its therapeutic effect in this model. In the setting of term HIE, however, TH does not provide full neuroprotection when it is deployed on its own, and infants can still suffer from debilitating long-term outcomes after an HI insult [37]. Part of the lack of complete neuroprotection by TH may be due to heterogeneous etiologies of injury in the clinical setting, including HIE complicated by infection or chronic placental insufficiency [38,39]. In particular, TH has not been neuroprotective in preclinical models of HI sensitized with LPS [13,40,41]. The lack of significant neuroprotection conferred by TH in our ferret model is therefore not unexpected due to the use of LPS for inflammatory presensitization, though TH animals did demonstrate improved behavioral outcomes in some tests compared to other HIH-exposed animals. For example, the TH group’s time to complete the reflex testing and activity level in the open field and MWM testing arenas were more consistent with the control group than the other HIH groups. However, gait analysis on the CatWalk showed that TH failed to provide behavioral recovery after HIH. Similar to the other HIH-exposed animals, the TH brains had shorter sulci, thinner gyri, and greater cerebellar exposure than the control animals, suggesting significant cortical dysgenesis after injury that was not ameliorated by TH. Additionally, and interestingly, despite evidence of diffuse white matter preservation on DTI after TH treatment, the TH group had the thinnest corpus callosum measurements out of all treatment groups. Though modeling complete chronic intra-uterine exposures such as chorioamnionitis remains a preclinical challenge, a significant number of infants still experience incomplete neuroprotection from TH treatment, and therefore models that respond similarly therefore remain important to the development of novel neurotherapeutics for scenarios where TH is not fully neuroprotective. 

Overall, exposure to HIH at P17 disrupted ferret locomotor function at P42, as seen through CatWalk, open field, and testing in a water maze arena. Ur failed to repair and appeared to significantly worsen these deficits. This aligns with our current findings for Ur across other behavioral and morphological parameters and further highlights Ur’s neuroprotective inadequacy in this model. Compared to animals in the Veh and Ur groups, Epo and TH animals tended to behave most like controls during late behavioral tests. Despite this apparent improvement, it is important to note that Epo and TH medians differed in some way from controls for most late motor parameters, indicating the persistence of neurobehavioral deficits in those animals despite any therapeutic signals. Considering the practical reliance on TH in the clinical setting, we find it necessary to highlight that, although TH animals showed potential improvement in some behavioral parameters, they often fell short on several behavioral parameters compared to Epo animals. 

This study has some limitations. Firstly, due to the lack of any therapeutic signal in the Ur group, these animals were not included in all of the final outcome analyses (particularly MRI and more extensive IHC measures). Regardless of the between-group differences observed across the behavioral parameters mentioned here, our findings are also more variable and nuanced compared to published studies using similar behavioral tests on rodents. Some of this may be due to the addition of confounding factors that arise when adapting rodent testing arenas to fit the larger body size of the ferret. The CatWalk system, in particular, seemed to confine and obstruct some ferrets from full ranges of motion. Few of the previous studies have examined motor function in juvenile ferrets beyond the development of reflexes [42]. Consequently, we have attempted to develop novel paradigms that seek to identify and understand the ferret-specific behavior we observed during testing at P42, with further development of testing paradigms including more complex cognitive function ongoing. A disadvantage of using the P17 ferret to model neonatal encephalopathies is that its human developmental correlate falls somewhere between late preterm and term gestational age, similar to the P7 rat [43]. This may help to explain why the partial neuroprotection from Epo seen in the model does not align with the lack of benefit from Epo treatment recently described in extremely preterm infants in the PENUT trial [44]. In its current state, the P17 model appears to more closely correlate with near-term HIE rather than preterm brain injury, though our future work in the model is planned to be either earlier (P12-~28 weeks’ gestation) or later (P21–full-term equivalent) to provide clear separation in developmental stage. It is worth mentioning that no sex differences in treatment response were observed in this study. However, there is a significant body of literature of both human and animal studies to support the hypothesis that there would be a difference in response to treatment observable in this model [45,46,47], and it is likely that sex-based differences would become apparent once larger group sizes are included. Another limitation is the relatively complex nature of the model, which was developed over several years of iteration [7]. As such, we are unable to know which of the individual components of the experimental protocol—LPS presensitization, hypoxia, hyperoxia, and reperfusion—drive a certain therapeutic response. Examining the mechanisms of Epo neuroprotection in this model are beyond the scope of this manuscript but will form the basis of future work. By comparison, although we have not studied response to these therapies without the presensitizing effect of LPS in the model, it is likely that the inclusion of LPS is the primary reason for a lack of neuroprotective effect of TH [13,40,41]. However, as more definitive evidence is accumulating regarding the lack of TH neuroprotection in lower income settings [5], perhaps due to more protracted or chronic perinatal insults, the model as used here may be particularly useful for examining therapies for use in scenarios where TH is not beneficial.

In summary, this model responds as expected to a known neurotherapeutic and provides a platform to model complex long-term cortical development and white matter pathology assessments not feasible in most rodent or other large animal models. The ability of translational research to bring new therapeutics to the bedside relies upon animal models that are able to faithfully replicate several aspects of the human disease state. We believe this model will help the field to achieve that goal.

## 4. Materials and Methods

### 4.1. Animals and Housing

All animal experiments were approved by the Institutional Animal Care and Use Committee at the University of Washington. Procedures were performed in accordance with the NIH Guide for the Care and Use of Laboratory Animals and reported in accordance with the ARRIVE (Animal Research: Reporting of In Vivo Experiments) guidelines. Ferret jills with cross-fostered and sex-balanced litters of 8 kits each were purchased from Marshall Bioresources (North Rose, NY, USA) and delivered to a centralized animal facility on postnatal (P) day 8. Ferrets were housed in a dedicated animal vivarium with ad libitum access to food and water before and after experimental procedures under standard conditions. Standard conditions included a 16:8 h light–dark cycle, room temperature range of 61–72 °F (16–22 °C), humidity of 30–70%, and 10–15 fresh air changes per hour.

### 4.2. Circulating Uridine and Uracil Levels

At P17, a single dose of uridine 5′ monophosphate (Sigma cat. U1752, St. Louis, MO, USA) was administered subcutaneously (s.c.) at a concentration of either 330 mg/kg (*n* = 5, low-dose Ur group) or 1000 mg/kg (*n* = 5, high-dose Ur group). Blood samples (150–200 μL; anticoagulated with EDTA) were obtained from the saphenous vein at 0 time point (baseline, *n* = 8) prior to the administration of Ur or saline vehicle. Subsequent blood samples were collected at 30 min, 1 h, 2 h, and 4 h. Circulating levels of uridine and uracil were measured using high pressure liquid chromatography-mass spectrometry (Appendix A). Our goal was to determine a Ur dose that resulted in circulating concentrations similar to those conferring neuroprotection in a rodent model of neonatal brain injury: 10,000 ng/μL [14].

### 4.3. Optimized P17 Injury Model

An overview of the experimental procedures is shown in Figure 6. At P17, ferret kits were removed from the nest, weighed, and randomized to either control or HIH conditions. Control animals and HIH ferrets received 3 mL/kg saline vehicle (Veh) or 3 mg/kg LPS (freshly prepared ULTRA PURE LPS from *E. coli* 055:B5, Lot #4231A, List Biological, Campbell, CA, USA) intraperitoneally, respectively. Inflammatory presensitizing injections were given approximately 4 h before the start of hypoxia to coincide with the peak cytokine response to LPS [40,48]. Animals who received LPS underwent bilateral carotid ligation under isoflurane anesthesia (3–5%) plus buprenorphine (0.05 mg/kg) analgesia. A 2 cm midline incision was made along the neck and the underlying tissue layers were dissected. Special care was taken to isolate the right and left carotid arteries from surrounding connective tissue and their accompanying nerves. The left carotid artery (LCA) was ligated twice with silk suture (5-0, Fine Science Tools, Foster City, CA, USA) and then transected between the two ligations. The right carotid artery (RCA) was temporarily occluded with umbilical tape (1/8 in, GF Health Products, Atlanta, GA, USA). The incision site was closed with wound clips, and the animals were taken to a separate room to recover. After a 30 min rest period, kits were placed in a prewarmed chamber and exposed to hypoxia (9% oxygen for 30 min) followed by hyperoxia (80% oxygen for 30 min), then hypoxia again (9% oxygen for 30 min). During the gas exposures, the internal body temperatures of two sentinel ferrets were closely monitored (Precision 4000A thermometer, YSI, Yellow Springs, OH, USA), to maintain normothermia (target rectal temperature 37 °C). After the second period of hypoxia, kits were returned to the surgical suite and the RCA umbilical tape was removed under anesthesia. In most cases, reperfusion of the RCA was observed. Incision sites were closed with wound clips and a 1 mL bolus of s.c. normal saline was given to maintain hydration. The kits were placed back with their jills to nurse and recover for 1 h.

### 4.4. Treatment

Initiation of treatment began 90–120 min after the end of the HIH insult across all cohorts. In the first cohort, HIH-exposed animals were randomized to receive either s.c. Veh or Ur (500 mg/kg every 12 h for 5 total doses). Ur dosing was determined based on pharmacokinetic data provided in Appendix A. In the second cohort, HIH-exposed animals were randomized to either Veh, Epo (2000 IU/kg s.c. at 0 h, 24 h, 48 h, and day 7 for 4 total doses), or TH (target rectal temperature 33.5 °C for 6 h). Epo dosing was based on previous rodent work in a similar model that conferred neuroprotection [49] and converted to an equivalent dose using the ferret’s body surface area [50]. Ferret kits not assigned to the TH group received their first dose of treatment (Veh, Ur, or Epo) and were placed in a plastic chamber in a heated water bath to maintain a normothermic rectal temperature of 36–37 °C. Appendix A shows normal nesting temperatures of a P17 ferret. Animals randomized to TH were placed in a separate water bath and water temperature was adjusted to maintain a rectal temperature of 33–34 °C. During the temperature management period, each animal had its rectal temperature measured every 15–30 min for 6 h.

### 4.5. Behavioral Testing

#### 4.5.1. Early Reflex Testing

Behavioral tests were performed daily from P21 to P27 and then 3×/week from P28 to P39. Kits were removed from the nest and placed in a heated chamber for 1 h prior to all testing. The tests were modified from rodent reflex testing and performed as previously published [7,11]. The tests included negative geotaxis (NG), cliff aversion (CA), and righting reflex (RR). NG was performed at both 25° and 45° and the time to rotate 90° and 180° (with a full step towards the top) was recorded. For CA, kits were positioned with their forepaws at the edge of the testing benchtop, and the times to back away from the edge and take a full step at least 90° away from the starting position were recorded. For RR, kits were positioned with their backs touching the testing benchtop. The experimenter would release the kit and the time it took to ground all 4 paws and the time to take a full step in any direction were recorded. All tests were performed by experimenters blinded to treatment groups. Kits underwent 3 trials each on NG and CA tests, and five trials each for the RR test. The animal was given a failing score if they were unable to complete the full test within 60 s or if they fell during NG or CA testing. A total time (sum of the mean time for each of the three tasks) score on each day, and the area under the curve (AUC) across the testing period (P21–P39) was calculated for each test to measure skill acquisition.

#### 4.5.2. Gait Analysis

Gait characteristics were assessed using the CatWalkXT system (Noldus Information Technology, Wageningen, Netherlands). The protocol for kit handling, experiment setup, and testing process has been previously published [11]. On P42, three compliant runs were collected from each animal. A compliant run was characterized by the ferrets’ ability to travel through the entire length of the testing platform without pausing. Paw print size, paw print intensity, base of support, phase dispersion, and lateral support were analyzed as measures of gait development. The software-detected paw print classifications were manually adjusted where needed to capture the ferret paw print more accurately, as the software was developed for use in rodents. In addition to the automated outputs, all runs were visually analyzed to determine the number of attempted runs. An attempted run was defined as the ferrets’ forward progression at least halfway through the CatWalk runway, regardless of full completion. A ratio of completed runs to attempted runs was calculated for each animal to measure animal compliance and ability to complete the trial.

#### 4.5.3. Open Field

After the CatWalk test on P42, the ferret’s behavior was observed in an open field. Animals were tested in an acrylic box measuring 55 × 55 cm with opaque walls measuring 63.5 cm in height. Using EthoVisionXT software (Stoelting Co., Wood Dale, IL, USA) and a video camera, the behavior and movement of the ferrets were recorded for 5 min. The outputs generated included total distance moved, velocity, time spent in the center, and time spent in the corners.

#### 4.5.4. Morris Water Maze (MWM)

A ferret-adapted version of the MWM test was performed on the final day of testing (P42) in a subset of the second cohort of animals (*n* = 76). Each animal was first given a 60 s acclimation period prior to the testing trials. The behavior and movement of each animal was filmed and tracked using EthoVision software. Each trial lasted up to 60 s or until the animal either mounted the escape platform or required rescue. Data collected during MWM were analyzed for frequency and velocity of movement and distance traveled. Aside from the computerized outputs, a secondary analysis was conducted by generating an image file of each animal’s complete swim path for every trial. The images were evaluated on a dichotomous scale for the following parameters: greater than 50% arena coverage and circuitous exploration in open water. A trial was deemed “circuitous exploration in open water” if it was a mainly non-linear swim path and if the ferret avoided the perimeter of the pool arena. An additional analysis calculated exploratory behavior as a function of distance traveled and time spent moving (see Appendix A).

### 4.6. Ex Vivo MRI

At P42, kits were euthanized and perfusion fixed with phosphate-buffered saline (PBS) followed by 10% neutral buffered formalin (NBF). Subsequently, the brains were removed and immersion-fixed in NBF for at least 72 h before being rinsed and submerged in PBS at 4 °C to rehydrate for a further 72 h. Brains were then mounted on agarose gel sleds inside 50 mL Falcon tubes and immersed in Fomblin (Perfluoropolyether, PFPE; Solvay Specialty Polymers, GA). Nuclear magnetic resonance (NMR) data were collected on the Bruker Avance III, 4.7 Tesla (200 MHz, 1H), 20 cm, horizontal bore magnet with ParaVision version 6.0.1 software. All settings were applied as described previously and provided in Appendix A [6]. Diffusion tensor images (DTIs) were obtained by processing diffusion-weighted images using the FMRIB Software Library (FSL v6.0, Oxford, UK) eddy software (https://fsl.fmrib.ox.ac.uk/fsl/fslwiki/eddy, accessed 4 January 2021) and FSL’s dtifit and fslmaths-fmedian options as previously detailed in Wood et al. [7]. The resultant FA maps were co-registered onto a generated template for tract-based spatial statistics (TBSS). Fractional anisotropy (FA) maps were presented using Threshold-Free Cluster Enhancement (TFCE) adjusted for multiple comparisons, with significance levels set at 0.05 and 0.15.

### 4.7. Brain Measurements

After MRI were complete, brains were rinsed with PBS to remove remaining Fomblin and placed back into NBF for a minimum of 48 h. The brain was then removed from NBF and placed onto a paper towel to absorb excess liquid. External cortical features were measured using digital calipers with fine pointed jaws (SRA Measurements Products, Walpole, MA, USA). Sulci were measured from the beginning and end of the most distinct portion of the corresponding sulcus. Gyri were measured from the widest aspect of each corresponding gyrus. A map of all brain measurements is shown in Appendix A. As measures of overall cortical development, all gyral and sulcal measurements for each animal were summed for comparison.

### 4.8. Immunohistochemistry (IHC)

After brain measurement, coronal slices at the level of the caudate nucleus were taken from each brain, embedded in paraffin, and 4 μm sections were prepared for hematoxylin and eosin (H&E) staining and IHC. Glial fibrillary acidic protein (GFAP, 1:300 dilution; Agilent (Dako)) and Iba-1 (1:1500 dilution, WAKO Chemicals USA, 019-19741) IHC was performed at the University of Washington Harborview Medical Center Histology IHC lab. Myelin basic protein (MBP, 1:500 dilution, Abcam, AB7349) and Oligodendrocyte transcription factor 2 (Olig2, 1:500 dilution, Millipore, AB9610) IHC was performed at the University of Washington Histology and Imaging Core. For GFAP, staining was performed using rabbit polyclonal anti-GFAP using the Leica Bond III IHC stainer, the polymer refine detection kit, and the Bond Epitope retrieval 1 solution for 20 min. Iba-1 slides were run on the same Bond autostainer platform using the Leica Bond refine detection kit. Heat epitope retrieval was used to unmask the antigen for 20 min in Leica Epitope Retrieval 1 solution. MBP slides were baked for 30 min at 60 °C and deparaffinized on the Leica Bond Automated Immunostainer (Leica Microsystems, Buffalo Grove, IL, USA). Antigen retrieval was performed by placing slides in EDTA for 20 min at 100 °C. The primary antibody in Leica Primary Antibody Diluent was applied for 30 min. A secondary antibody, unconjugated rabbit anti-rat IgG (1:300 + 5% NGS in TBS, Vector, AI-4001), was then applied for 8 min. Goat anti rabbit horseradish peroxidase Leica Bond Polymer was added for 8 min. Antibody complexes were visualized using Leica Bond Mixed Refine (DAB, 3,3′-diaminobenzidine) detection 2× for 10 min at room temperature (RT). For Olig2, staining was performed using rabbit polyclonal anti-Olig2 (Olig2; Millipore, Cat No. AB9610) on formalin-fixed paraffin-embedded sections. Antigen retrieval was performed by placing slides in Citrate for 20 min at 100 °C. The primary antibody, Olig2 (1:500) or Rabbit IgG (1:1000) in Leica Primary antibody diluent, was applied for 30 min at room temperature.

### 4.9. Quantitative IHC

Image analysis was performed using whole slide digital images and automated image analysis. Slides were scanned in bright field with a 20× objective using a Nanozoomer Digital Pathology slide scanner (Hamamatsu; Bridgewater, NJ, USA) and the digital images were imported into Visiopharm software (Hoersholm, Denmark) for quantitative analysis. Prior to analysis, regions of interest (ROIs) were manually traced on each image for each stain. The ROIs were traced following a standardized protocol on both the left and right hemisphere and included the thalamus, subcortical white matter (SWM), dorsal cortex, hippocampus, and the corpus callosum (CC). The software then converted the initial digital imaging into gray scale values using two features, RGB-R with a mean filter of 5 pixels by 5 pixels and an RGB-B feature. Visiopharm was then trained to label positive staining and the background tissue counter stain using a stain-specific configuration based on threshold pixel values. Images were processed in batch mode using this configuration to generate positively stained to unstained tissue ratios. The research staff were blinded to the treatment groups during all assessments. 

Using GFAP-stained slides, the thickness of white matter in the sub-sulcal regions of the SWM was measured. The images were imported into NDP.view2 (Hamamatsu Photonics, Bridgewater, NJ, USA) and measured using the annotation tool. White matter thickness was assessed in 4 ROIs: the CC and three white matter regions inferomedial to the lateral sulci, suprasylvian sulci, and pseudosylvian sulci. 

MBP-stained slices were also used to measure the area of SWM in the control, Veh, and TH groups. The images were exported from NDP.view2 at 0.63× after adjusting the image settings to sharpen and lower gamma correction (0.5), which darkened and clarified the white matter tracts. The images were then imported into ImageJ, where the freehand selection tool was used to trace three subcortical ROIs-lateral, suprasylvian, and pseudosylvian-named for the corresponding sulcus perpendicular to each band of white matter (Appendix A).

### 4.10. Cortical Neuropathology Scoring

The H&E-stained slides were evaluated by a board-certified veterinary pathologist (JMS), who was blinded to treatment group and scored for cortical lesion and mineralization on a scale of 0 to 4. Cortical lesions were scored based on modifications of previous scoring systems used in rat models of hypoxic-ischemic brain injury [51,52] as follows: 0 = no detectable lesion; 1 = small focal or multifocal (<3) area of neuronal necrosis/loss and/or rarefaction, comprising <10% of the cortex; 2 = regionally extensive or multifocal lesion with neuronal necrosis/loss and/or rarefaction/cavitation, affecting 10–33% of the cortex unilaterally or 5–25% bilaterally; 3 = multifocal to coalescing lesion affecting 33–66% of the cortex unilaterally or 26–50% bilaterally; and 4 = severe lesion with cavitation affecting >66% of the cortex unilaterally or >50% bilaterally. Mineralization, which has been previously described following neuronal degeneration in animal models of transient forebrain ischemia [53], was scored as follows: 0 = no detectable lesion; 1 = up to two small regions of mineralization; 2 = multifocal mineralization, unilateral or bilateral (3–10 foci); 3 = regionally extensive and multifocal to coalescing mineralization, bilateral, >10 foci; and 4 = severe bilateral multifocal to coalescing mineralization with thalamic involvement to any extent. Mineralization was characterized histologically by basophilic punctate, granular material on H&E and confirmed in an affected animal with von Kossa staining.

### 4.11. Statistical Analysis

In order to directly compare the Ur and Epo/TH cohorts but account for inter-cohort variability, data from behavioral testing and brain measurements were adjusted relative to the median control animal value of that sex for that cohort. For cortical neuropathology scores, animals that died were included and given a score of one increment greater than the worst possible score, to allow for assessment of combined outcomes. Data across multiple groups were compared using a Kruskal–Wallace test with Dunn’s post hoc test for pairwise tests adjusted for multiple comparisons. Data are presented as median with interquartile range (IQR). Adjusted *p*-values < 0.05 were considered statistically significant. Statistical analyses were performed in Prism (GraphPad Software, San Diego, CA, USA) and R Version 3.6.3 (Vienna, Austria).

## Figures and Tables

**Figure 1 ijms-22-09841-f001:**
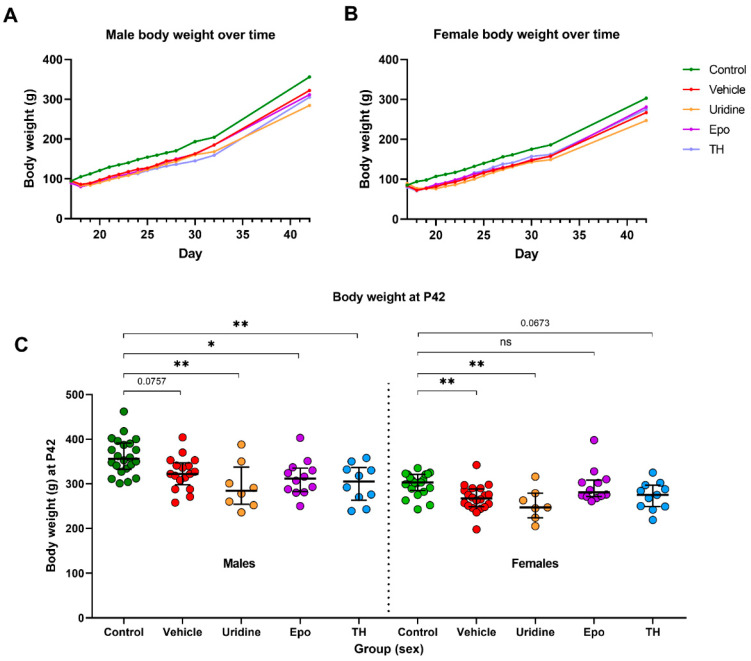
Body weight analysis. (**A**) At P42, all Ur animals were significantly lighter than control animals. In the males, the other HIH animals were also lighter than the controls; however, the comparison between control males and Veh males was not significant. In the females, the Veh animals were significantly lighter than the control animals, but TH and Epo animals were more comparable to the control animals. (**B**) On the day of HIH (P17), mean (SD) weight of the male kits was 93.0 (7.8) g. On average, HIH males lost 9.6% (7.1%) of their body weight between P17 and P18 as a result of the HIH insult. By P19, HIH males gained 2.0% (6.9%) of their P17 body weight back, compared to a 26.7% (6.0%) weight gain in controls by P19. The HIH males never fully recovered from the weight loss that followed HIH, but the difference in weight between Veh males and control males on P42 lost its significance. (**C**) At P17, the mean weight of the female kits was 83.3 (5.6) g. On average, HIH females lost 11.3% (8.3%) of their body weight between P17 and P18 as a result of the HIH insult. By P20, HIH females gained 4.4% (13.5%) of their P17 body weight back, compared to a 25.2% (9.0%) weight gain in controls by P20. The HIH females took an extra day longer to gain back the P17 body weight deficit when compared to the HIH males, but the HIH females maintained a weight gain pattern more consistent with the control females by P42. * Denotes *p*–value < 0.05, ** denotes *p*–value < 0.01, and ns denotes no significant difference.

**Figure 2 ijms-22-09841-f002:**
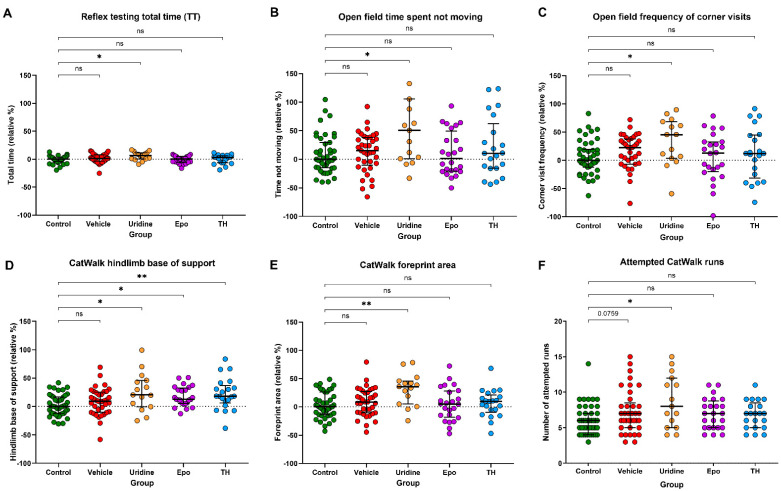
Motor development. (**A**) AUC analysis of the total time (median; IQR) to complete all reflex tests showed that Ur animals (8263.3 s · days; 7782.8–8658.0 s · days) took significantly longer to complete reflex testing compared to control animals (8083.3 s · days; 7690.5–8486.2 s · days) throughout the testing period. (**B**) Frequency of open field corner visits in the Veh, Ur, Epo, and TH groups relative to the control group. Median corner visit frequency in controls was 49. Median values were significantly increased in the Ur group by 45.5% and non–significantly increased in all other HIH–exposed groups relative to control median. (**C**) Time in seconds (s) spent not moving in open field in the Veh, Ur, Epo, and TH groups relative to the control group. Median time spent not moving was 99.7 s in the control group and significantly increased in the Ur group by 50.7%. In all other HIH groups, median time spent not moving was non–significantly increased. (**D**) Hindlimb BOS in the Veh, Ur, Epo, and TH relative to the control median. Median hindlimb BOS was 0.01 cm/g in the control group and was significantly increased in the Ur, Epo, and TH groups by 20.3, 12.5, and 18.0%. (**E**) Foreprint area in the Veh, Ur, Epo, and TH groups relative to the control median. Median foreprint area was 0.39 cm/g in the control group and significantly increased by 35.6% in the Ur group. In all other HIH–exposed groups, median foreprint area was non–significantly increased. (**F**) Number of attempted runs in the control, Veh, Ur, Epo, and TH groups. The median number of attempted runs was 6 in the control group and significantly increased to 8 in the Ur group. All other HIH groups had non–significant increases in number of attempted runs. * Denotes *p*–value < 0.05, ** denotes *p*–value < 0.01, and ns denotes no significant difference.

**Figure 3 ijms-22-09841-f003:**
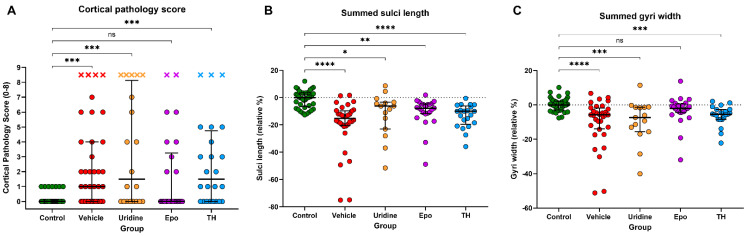
Cortical Pathology and Morphology. The median and IQR is plotted on each graph. (**A**) Surviving animals received combined scores of 0–8 (0–4 cortical pathology + 0–4 mineralization) while animals that died after the HIH insult received a score of 8.5. Control animals had a median cortical pathology score of 0 with no deaths. Median scores significantly increased in the Veh group to 1 and in the Ur and TH groups to 1.5. Deaths also increased in the Veh, Ur, and TH groups to 4, 5, and 3, respectively. Compared to controls, number of deaths in the Epo group increased to 2. (**B**) Summed sulci length in the HIH groups relative to the control group. After calculating the sum of the lengths of each sulci, the HIH groups all displayed a shorter sum of sulcal length than the control group. On average, HIH brains sulci were 8.8% (3.3–16.9%) shorter than control brains. (**C**) Summed gyri widths in the HIH–exposed groups relative to control median. Similar to the summing of sulci, the summing of the width of each gyri showed that control brains had significantly wider gyri than Veh (3.6%; 0.1–9.5%), Ur (7.3%; 1.4–15.6%) and TH (5.4%; 2.6–8.4%) brains. Epo brains exhibited summed gyral widths more similar to control brains. * Denotes *p*–value < 0.05, ** denotes *p*–value < 0.01, *** denotes *p*–value < 0.001, **** denotes *p*–value < 0.0001, and ns denotes no significant difference.

**Figure 4 ijms-22-09841-f004:**
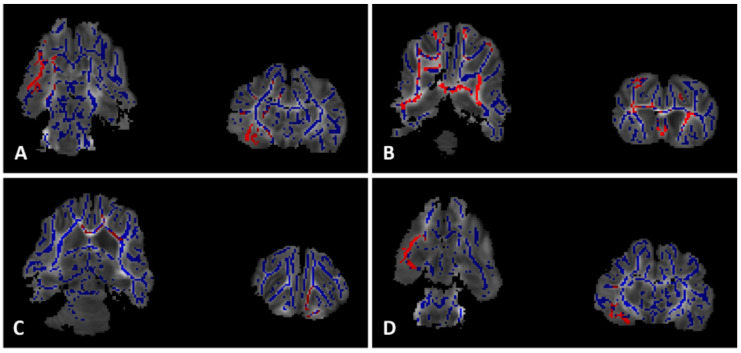
Diffusion Tensor Imaging. Comparison of FA values throughout the white matter (blue). Light red indicates *p* < 0.15 and dark red indicates *p* < 0.05 after threshold-free cluster enhancement (TFCE) adjustment for multiple comparisons. (**A**) Contol FA values greater than Veh FA values in the dorsal striatum and substantia innominata. (**B**) TH FA values greater than Veh FA values throughout cerebral white matter. (**C**) Epo FA values greater than Veh FA values in the anterior cerebral white matter and interior capsule. (**D**) Control FA values greater than Epo FA values in the peri-hippocampal cerebral white matter tracts.

**Figure 5 ijms-22-09841-f005:**
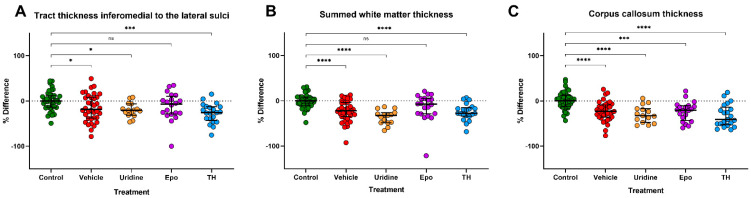
Quantitative IHC. (**A**) After measuring the subcortical white matter (SWM) tract thickness of the area inferomedial to the lateral sulcus in the GFAP–stained slices, the length of the WM tracts (median % difference; IQR) was significantly decreased in Veh (15.4%; 5.1–31.8%), Ur (20.2%; 6.3–31.3%), and TH (25.6%; 12.5–42.8%) brains when compared to controls. (**B**) In the three regions of interest (ROIs) in the white matter measured, the summed measurements of tract thickness were significantly thinner in Veh (15.6%; 1.6–30.0%), Ur (32.0%; 26.1–48.3%), and TH (27.6%; 15.2–33.6%) brains when compared to control brains. Epo brains had white matter thickness more consistent with controls in all ROIs. (**C**) Compared to controls, the corpus callosums (CC) of the Veh (21.4%; 3.9–40.6%), Ur (32.6%; 17.0–47.3%), Epo (25.1%; 12.7–44.9%), and TH (41.0%; 13.6–52.3%) brains were significantly thinner when compared to the CC of control brains. * Denotes *p*–value < 0.05, *** denotes *p*–value < 0.001, **** denotes *p*–value < 0.0001, and ns denotes no significant difference.

**Figure 6 ijms-22-09841-f006:**
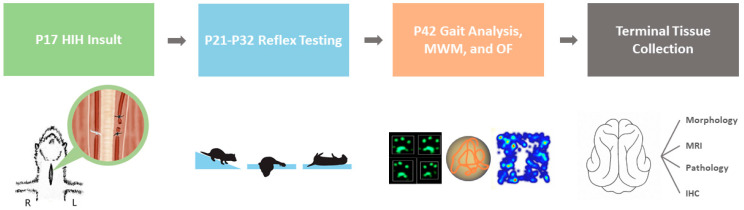
P17 Model Timeline. On P17, animals were administered 3 mg/kg LPS before undergoing bilateral carotid artery ligation, with the RCA occlusion being reversed after the gas exposure (hypoxia-hyperoxia-hypoxia). Special care was taken to isolate the artery away from surrounding vessels and nerves. Treatment was then initiated within a 90–120 min time window after the end of the HIH insult. Basic reflexes were assessed from P21 to P32 followed by a battery of late behavioral testing on P42. The brains were then extracted and processed for further evaluation.

## Data Availability

The data presented in this manuscript are available from the corresponding author upon reasonable request.

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
