# Peer review of "Evaluating Neuroprotective Effects of Uridine, Erythropoietin, and Therapeutic Hypothermia in a Ferret Model of Inflammation-Sensitized Hypoxic-Ischemic Encephalopathy"

_ijms, 2021, doi:10.3390/ijms22189841_

Round 1

Reviewer 1 Report

The manuscript by Corry et al. „Evaluating Neuroprotective Effects of Uridine, Erythropoietin, and Therapeutic Hypothermia in a Ferret Model of Neonatal Encephalopathy“ uses a pre-described neonatal late preterm animal model. The authors have published the description of the model previously and are one of the leading research groups in the field. The research question is of major importance as therapeutic hypothermia lacks neuroprotection in more than 30% of asphyxiated newborns and therefore additional treatments are urgently needed. The manuscript evaluated two treatments in comparison with cooling, being Uridine and Erythropoietin. The research group has large pre-clinical and clinical experience with Erythropoietin. The authors did find meaningful neuroprotection by Uridine and therapeutic hypothermia but a trend towards protection for Erythropoietin. This finding is of major interest and will lead to further studies in this field.

The authors used different readouts, including neurofunctional testings, which is very important for the findings.

I have some questions/comments, which will hopefully further improve the quality of this important manuscript:

  1. Title: Why do the authors not say “inflammation-sensitized hypoxic-ischemic encephalopathy” but “neonatal encephalopathy”? I would suggest to change the title, as this describes the nature of the model. Neonatal encephalopathy may be caused by many other underlying diseases (e.g. metabolic, genetic,…).
  2. Introduction line 39: maybe the authors want to add the just published HELIX-trial (Thayyil S et al., Lancet Glob Health. 2021 Aug 3;S2214-109X(21)00264-3)
  3. Line 56: This is not really clear to me. Was the hypothesis that Ur and Epo would provide significant neuroprotection against HI or LPS-HI?
  4. Model: the animal model appears quite complex with a mixture of bilateral carotid occlusion, hypoxia and hyperoxia, with and without LPS. This has been described and published previously – however, what do the authors think does this model reflect to in the clinical scenario?
  5. What does the P17 ferret brain correspond to from a maturation point of view (which week of gestation in a human)?
  6. The LPS dose appears quite high. Have the authors used the same LOT and batch of LPS for each experiment? Have the authors seen any difference in LPS administration?
  7. Is there any reason why Uridine was administered s.c. instead of i.p.?
  8. Of interest: would it be possible to monitor the animals with aEEG or EEG, as in other large animal models during the insult? Were any hemodynamic parameters monitored during the insult? Would that be feasible for the future?
  9. What is the normal nesting temperature of a P17 ferret? Have the authors ever tested other temperatures for TH – as they did in rats (Wood et al, Sci Rep. 2016 Mar 21;6:23430)?
  10. Line 459: was the 500mg/kg Ur based on the findings of the experiments performed under chapter 4.2? If so, please state that here. Why was Uridine administered every 12h and not 24h as in the rodent model?
  11. Line 460: why did the authors use 2000IU/kg Epo?
  12. Line 509: I suppose the description of the Morris Water Maze is missing. Please add.
  13. Were all IHC analysis performed at P42 brains? Would the authors suggest to observe changes at earlier time points after the insult? What is the time frame – from a maturation point of view – between P17 and P42 in a ferret brain?
  14. Of interest: do the authors always see standard bilateral brain injury in this model, or is the injury more severe on the LCA hemisphere?
  15. Has the cortical neuropathology scoring been evaluated before?
  16. Statistics: did the authors include all dead animals in the analysis? This would appear strange, as animals might have died at different time points. Please clarify. Did the authors analyse the effect of gender, weight, experimental date (first set vs second set of experiments) on outcome?
  17. Results: line 70: correct Outsomes to Outcomes please.
  18. It would be good to include an experimental flow chart so that the reader can follow which animal was used for what and to follow which animal died when in which group.
  19. Why did the authors use such large group sizes?
  20. Why was the mortality so high in the Ur group?
  21. Line 102 – this statement is correct, however the HIH animals were non-significantly slower in all tasks, right? If so, please state that here.
  22. Line 163: this is an important finding. MWM seems to be not the right outcome test in the ferret model. This is important for all future research with this model. The readout finding should be interpreted with caution – as stated by the authors.
  23. How would the neuropathology findings look like without the dead animals? I am not sure if including these animals is right, as the pathology was performed at P42 and will be different from P17 ferrets.
  24. Discussion: have the authors ever used this model without LPS pre-sensistization showing neuroprotection with hypothermia?
  25. Limitations of the study are clearly stated in the manuscript

Reviewer 2 Report

General Comments

This study uses a newly developed ferret model to simulate brain injury in premature infants. The model consists of pre-sensitizing inflammation (LPS), carotid ligation, reversible carotid occlusion, hypoxia and hyperoxia in an effort to simulate most of the potentially known factors that predispose to brain injury in premature infants. The authors then attempt to employ uridine monophosphate, erythropoietin, and hypothermia in an effort to ameliorate injury to the brain. The advantage of the ferret is that it is larger than smaller rodents providing more brain tissue and has a gyrencephalic brain in comparison to smaller rodents such as mice and rats, which have lissencephalic brains. It appears that it must be somewhat difficult to induce brain injury in this model as the authors are employing multiple strategies in their model. Therefore, the question arises as to which maneuver induces which aspect of injury and, hence, which effects of the neuroprotectants? Disappointingly, none of the strategies employed showed much neuroprotection. The strength of the manuscript is that the authors have included MRI, behavioral and neuropathic changes in their analysis. Another merit is the large number of animals included. The weaknesses are the multiple injuries limit the ability to know what caused each of the injuries and how the partial neuroprotection attenuated the injuries. Hence, the mechanisms of injury are difficult to discern. Please see additional specific comments below.

Specific Comments

  1. Erythropoietin has recently been shown not to be efficacious in protecting the premature brain. Therefore, it is not surprising that it is also not very protective in this model.
  2. When studying animal models and time of injury it is necessary to adjust time for the length of the animal life. How does the life span of the ferret relate to the lifespan of the human? Is 6 hours of the ferret life equivalent to the 6 hours in a human baby?
  3. The mortality rate in this model is quit high which could affect the outcomes as acknowledged by the authors.
  4. 3 Scatterplot
  5. Line 303 represents speculation.

Author Response

Authors’ response: We are very grateful to the reviewer for their comments and have tried to better acknowledge the limitations of this complex model. Further point-by-point responses and resulting edits are provided below.

Please see additional specific comments below.

Specific Comments

  1. Erythropoietin has recently been shown not to be efficacious in protecting the premature brain. Therefore, it is not surprising that it is also not very protective in this model.

Authors’ response: Any neuroprotective effect of Epo in preterm infants in the clinical setting remains a question that is complex to answer as it relies on other factors, such as iron status (German et al., Journal of Pediatrics 2021; 10.1016/j.jpeds.2021.07.019). However, it’s likely that treatment with Epo alone in this and other models will never provide full neuroprotection. As part of future studies, we’d like to explore adjunctive therapies that will expand upon the modest neuroprotection conferred by Epo in this model.

  1. When studying animal models and time of injury it is necessary to adjust time for the length of the animal life. How does the life span of the ferret relate to the lifespan of the human? Is 6 hours of the ferret life equivalent to the 6 hours in a human baby?

Authors’ response: It’s difficult to compare development between species in hours as certain cellular process may occur at relatively fixed timescales; however, 6 hours of the ferret life roughly translates to 12-24hrs of human life.

  1. The mortality rate in this model is quite high which could affect the outcomes as acknowledged by the authors.

Authors’ response: We agree, though baseline mortality rates around 10% are not unexpected in similar models in other species, which is a function of the severity of insult required to provide reproducible injury. Unfortunately in this study, we saw compounding mortality from Ur administration that was unexpected.

  1. 3 Scatterplot

Authors’ response: We apologise that we weren’t certain exactly certain what this comment referred to, though the legend for Figure 3 has been edited to increase accuracy by removing mention of “scatter plots”. (Line 200)

  1. Line 303 represents speculation.

Authors’ response: Our PK data does suggest that the ferret does convert most of the circulating uridine to uracil. And as other groups are still investigating the neuroprotective effects of uridine for neonatal brain injury, we feel we are justified in not making a blanket statement on its potential usefulness in other models or in humans.

Reviewer 3 Report

The manuscript submitted by Kylie A. Corry et al., entitled "Evaluating Neuroprotective Effects of Uridine, Erythropoietin, and Therapeutic Hypothermia in a Ferret Model of Neonatal Encephalopathy" is highly interesting. The manuscript is original, well written and structured. Differently by previous studies present in literature this report that Therapeutic hypothermia in sinergy with Eritropoietin treatment appear to provide significant, though not complete, neuroprotection in this mouse model. The results are well described and support the conclusions of the study. I propose the acceptance in the present form of this manuscript.

Author Response

Authors’ response: We are very grateful to the reviewer for their positive comments, and we hope that they feel the manuscript has benefitted from the revisions.

Reviewer 4 Report

The paper presents the results of an animal study evaluating the responds to neurotherapeutic agents and it provides a platform to model complex long-term cortical development and white matter pathology assessments. The study points out the ability of translational research to validate new therapeutics for the clinical use, relying on animal models. The study is valuable, original and with high scientific level.

Author Response

Authors’ response: We are very grateful to the reviewer for their positive comments.

Reviewer 5 Report

The paper written by the following Authors: Kylie A. Corry, Olivia R. White, AnnaMarie E. Shearlock, Daniel H. Moralejo, Janessa B. Law, Jessica M. Snyder, Sandra E. Juul and Thomas R. Wood, entitled “Evaluating Neuroprotective Effects of Uridine, Erythropoietin, 2 and Therapeutic Hypothermia in a Ferret Model of Neonatal 3 Encephalopathy” presents an interesting study with the following hypothesis “Ur and Epo would provide 56 significant neuroprotection, but that LPS pre-sensitization would negate the protective 57 effect of TH”.

Although the paper is interesting, I have some major concerns:

Title

The title reflects the results presented here.

Abstract

The abstract is lacking the aim, the material and methods description as well as an informative conclusion. It should be written in more details.

Material and Methods

  1. Three is no information about the number of analyzed objects. It should be included in the manuscript.
  2. There is no information about the tool for measurement of animal’s temperature. What was its accuracy?

Discussion

In the discussion part there is no limitation to the studies. It should be included in the manuscript.

Author Response

Authors’ response: We are very grateful to the reviewer for their comments and suggestions. Point-by-point responses and resulting edits are provided below.

Although the paper is interesting, I have some major concerns:

Title

The title reflects the results presented here.

Abstract

The abstract is lacking the aim, the material and methods description as well as an informative conclusion. It should be written in more details.

Authors’ response: The abstract has been edited to make the aim clearer (Lines 14, 17-18), but we feel that the methods and conclusions are reasonable and informative given the journal’s word limits.

Material and Methods

  1. There is no information about the number of analyzed objects. It should be included in the manuscript.

Authors’ response: Under the Model Outcomes section, we present the group totals included in the manuscript. At the suggestion of Reviewer 1, we’ve added a figure to the supplemental materials that shows animals numbers pre- and post-treatment assignment (Supplemental Figure 2), as well as which animals survived to the full assessment of outcomes at P42. We hope this clarifies the number of analyzed objects.

  1. There is no information about the tool for measurement of animal’s temperature. What was its accuracy?

Authors’ response: We apologize for this omission. All rectal temperatures were measured using a YSI Precision 4000A Thermometer with (accuracy within 0.01°C). These details have been added to the methods section. (Line 486)

Discussion

In the discussion part there is no limitation to the studies. It should be included in the manuscript.

Authors’ response: We have expanded the limitations in the discussion based on suggestions from Reviewer 1. (Line 416-420)

Round 2

Reviewer 2 Report

The manuscript has been improved by the revisions. However, this reviewer still has concerns that with such a complex model, one cannot tell what is causing what abnormality or how the neuroprotective agents are functioning. Why is it so hard to cause brain damage in this model?

Author Response

The manuscript has been improved by the revisions. However, this reviewer still has concerns that with such a complex model, one cannot tell what is causing what abnormality or how the neuroprotective agents are functioning. Why is it so hard to cause brain damage in this model?

Authors’ response: We do acknowledge the complexity of the model, which does complicate the examination of mechanistic pathways involved in the action of specific neurotherapeutics. Though the model is complex, we do feel that we have been realistic about it's development and limitations. Previous publications have outlined the iterative process of developing the current model (Wood et al., Developmental Neuroscience 2018, Supplementary Table 1; https://doi.org/10.1159/000498968), and acknowledged that, as with any preclinical model, the goal was not accurately reproduce the exposures encountered by infants clinically, but to provide a confluence of the mechanistic factors thought to be involved in neonatal hypoxic-ischemic brain injury including inflammation, hypoxia, and oxidative stress (Wood et al., JoVE 2019; https://dx.doi.org/10.3791/60131).

Additional wording has been added (lines 399 and 402) to more directly state that examining the mechanisms of neuroprotection are beyond the scope of this manuscript, though we do believe that it is fair to say that the model responds as would be expected given related research in the field and the factors involved. We are also actively investigating factors that contribute to the apparent resilience to injury in this model.

Reviewer 5 Report

I accept the manuscript in present form.

Author Response

Authors’ response: We are very grateful to the reviewer for taking the time to re-review our manuscript.